# Qualitative and Visual Along-Tract Analysis of Diffusion-Based Parameters in Patients with Diffuse Gliomas

**DOI:** 10.3390/brainsci14030213

**Published:** 2024-02-26

**Authors:** Markus Fahlström, Sadia Mirza, Åsa Alberius Munkhammar, Maria Zetterling, Francesco Latini

**Affiliations:** 1Department of Surgical Sciences, Molecular Imaging and Medical Physics, Uppsala University, 75185 Uppsala, Sweden; 2Department of Medical Sciences, Section of Neurosurgery, Uppsala University, 75185 Uppsala, Sweden; 3Department of Rehabilitation and Pain Centre, Uppsala University Hospital, 75185 Uppsala, Sweden

**Keywords:** white matter, diffuse gliomas, along-tract analysis, DTI, neuropsychological impairment

## Abstract

Background: Grade 2–3 diffuse gliomas (DGs) show extensive infiltration through white matter (WM) tracts. Along-tract analysis of WM tracts based on diffusion tensor tractography (DTI) can been performed to assess the microstructural integrity of WM tracts. The clinical implication of these DTI-related findings is still under debate, especially in tumor patients. The aim of this study was to analyze and compare diffusion-based parameters along WM tracts and variables specific to WM -tumor interactions in DGs and correlate them with preoperative neuropsychological assessment. Methods: Fourteen patients with IDH-mutated grade 2–3 DGs were included. Tumor volumes were manually segmented on 3D-FLAIR images after spatial normalisation to MNI space. DTI was acquired using a single-shot echo-planar sequence on a 3T with 48 sampling directions. DTI data were reconstructed within the MNI space using q-space diffeomorphic reconstruction (QSDR) in DSI studio. Five bilateral sets of WM tracts were reconstructed based on the HCP-1065 template. All WM tracts were stretched to the same length of 100 indices, and for each index diffusion-based parameters fractional anisotropy (FA), radial diffusivity (RD), axial diffusivity (AD), mean diffusivity (MD) and quantitative anisotropy (QA) were sampled. Tumor-related parameters (TRP); tumor volume (Tv), maximum tumor presence (MTP) and the number of sequential indices in which a tumor is present (Te) were derived based on the along-tract analysis. Normal data were constructed by calculating the average and standard deviations of contralateral and not-affected WM tracts for each diffusion-based parameter, respectively. Affected WM tracts were individually compared to normal data using a z-test. Preoperative neuropsychological assessment was performed in all subjects and correlated to results from the along-tract analysis using correlation and logistic regression models. Results: Abnormalities in diffusion-based parameters were detected in WM tracts. Topographical and quantitative information were presented within the same graph. AD and MD displayed the highest linear correlation with the TRPs. Abnormal QA showed a linear correlation with Tv per WM tract. Neuropsychological impairment was correlated with all the TRPs and with abnormal FA (*p* < 0.05) and abnormal QA (*p* < 0.01). Abnormal QA was the only independent variable able to predict the presence of neuropsychological impairment in the patients based on the linear regression analysis. Conclusions: Graphical presentation of the along-tract analysis presented in this study shows that it may be a sensitive and robust method to acquire and display topographical and qualitative information regarding WM tracts in close proximity to DGs. Further studies and refinements to the methods presented herein may advance current clinical methods for evaluating displacement and infiltrations and further aid the efforts of pre-planning surgical interventions with the goal to maximise EOR and tailor oncological treatment.

## 1. Introduction

Diffuse gliomas (DGs) (WHO 2 and 3) are primary brain tumors derived from glial cells. They occur mainly in adult life with a peak incidence around 30–40 years [1,2]. The modern classification of a DG is based on molecular features such as IDH mutation status and 1p19q codeletion which are currently applied in treatment-planning decisions [3]. An advanced understanding of the metabolic effects induced by IDH mutations offers opportunities for specific targeted therapies that may improve patient outcomes [4,5]. DGs are characterized by a relatively slow natural course but extensive and continuous infiltration of the host brain [2,6]. They tend to preferentially infiltrate “secondary” functional areas (immediately near the so-called primary eloquent regions), or the so-called “minimal common brain” [7,8,9]. These specific tumor features impose a major challenge on their clinical management, requiring an individualized approach for each patient to decide the optimal treatment strategies [2]. It has been demonstrated that the tumor tends to interfere with normal brain function by disrupting the functional connectivity of brain networks within the peritumoral and distant brain areas, thereby inducing neuropsychological impairment and/or seizure activity [2,10,11,12,13]. The mechanisms of cortical–subcortical plasticity at the individual level, with important anatomical-functional variability, are not fully understood [13,14,15]. Surgical resection is, however, the first treatment strategy in which the extent of resection (EOR) correlates well with prolonged survival [16,17,18].

Despite the advances in neurophysiological and high-order functions monitoring during surgery, the surgical results and the postoperative patient status still depend on the initial tumor location and their infiltration into large-scale networks [6,11,17]. An extensive preoperative assessment including language and neuropsychology and cognitive functions is important to define the involvement of white matter networks at the diagnosis and to detect signs of tumor-induced neuroplasticity [13,19,20]. Moreover, this information may guide the neurosurgeon to better understand the individual connectome of the patient and to tailor the intraoperative mapping based on the complex dynamic interaction expected with the surrounding brain [21,22]. In fact, the subcortical white matter (WM) tracts often represent the main limit to a surgical resection and, at the same time, they represent the way of least resistance that DG cells use to disseminate [23,24,25]. A better comprehension of the WMs interaction with DGs has become pivotal in several fields of neurosurgical oncology to improve diagnosis, to detect treatment response and to predict outcomes.

Among the major advances in terms of preoperative imaging, tractography is the most widely established approach [26,27,28,29]. Tractography, established on magnetic resonance-based diffusion tensor imaging (DTI) has been utilized as a tool for the three-dimensional visual evaluation of WM tracts. This method offers the possibility to assess displacement, infiltration and disruption from DGs, aiding the neurosurgeon during resection to improve the resection rate and preserve postoperative functionality [26,28]. In addition, DTI has also been applied to indirectly and quantitatively measure the microstructural integrity of white matter, providing different information about WM, axonal or myelin integrity [30,31,32,33,34]. Diffusion-based metrics such as fractional anisotropy (FA), radial diffusivity (RD), axial diffusivity (AD) and mean diffusivity (MD) are currently used as measures of tissue microstructure, detecting WM abnormalities. Quantitative anisotropy (QA) is also used as a measure of anisotropy within a diffusion process in a biological tissue. It is described as being less affected by edema, while FA and AD are also sensitive to edema and are used in creating high-definition diffusion data [27,35,36].

On the other hand, these measures are inherently nonspecific and may depend on several possible biological mechanisms underlying WM microstructural modifications [37,38]. A correlation between specific WM tracts and neuropsychological tests have been described using DTI and intraoperative tests [15,39]. The majority of these studies use the analysis of the average of metrics over all voxels of the tract, considering the bundle as a whole. WM tract averages involve the calculation of statistics over the entire WM tract, which can be useful for investigating global changes in WM tract integrity. However, it does not provide information about regional variations along the tract, which are, for example, more relevant in the case of tumor infiltration [23,40]. Tract-specific analysis has evolved during the last 15 years from studying the averages of diffusion-based parameters for each tract [41,42,43] to an analysis of microstructural parameters in multiple segments along the tract [44,45].

Along-tract analysis involves dividing the WM tract into smaller segments of arbitrary length along its course and analyzing the diffusion-based parameters in each segment separately [46]. This method provides a more detailed picture of variations along the tract, allowing for a graphical presentation which can visualize alterations and specific functions associated with different segments of the WM tract. Along-tract analysis has previously been used in a variety of applications; however, the results that have been published warrant further exploration into accuracy and limitations [37,43,46,47,48,49,50,51,52,53,54,55].

The aim of this study was to qualitatively and visually assess, by graphical presentation, different Diffusion-based parameters along tracts in patients with DGs and to evaluate any potential relationships with preoperative neuropsychological assessment.

## 2. Materials and Methods

### 2.1. Patient Population

Patients (>18 years) presenting with a radiological diagnosis of suspected DLGGs were consecutively recruited at the Department of Neurosurgery, Uppsala University Hospital, Uppsala, Sweden, and enrolled as a part of larger study between August 2014 and August 2022. Fourteen patients from the larger cohort that underwent a surgical resection between 2020 and 2022, and who had a confirmed diagnosis of a grade 2–3 DG, were analyzed in the current study. Exclusion criteria were previous resection for brain tumors, previous radio-chemotherapy and background of psychiatric or severe cognitive impairment or impossibility to perform neuropsychological assessment due to medical constraints. The study was approved by the institutional ethics review board (Dnr 2015-210-2 and Dnr 2023-00876-01). Informed consent was obtained prior to surgery at the Department of Neurosurgery, Uppsala University Hospital, Uppsala, Sweden.

### 2.2. Imaging

DTI was acquired using a single-shot echo-planar sequence on a 3.0 Tesla MR scanner (Achieva, Philips Healthcare, Best, The Netherlands) with 48 sampling directions and a b-value of 1000 s/m^2^. A total of 60 axial slices were acquired with an in-plane resolution of 1.75 mm and slice thickness 2 mm. We acquired 3D-T2 fluid attenuated inversion recovery (T2-FLAIR) images for morphological evaluation and manual tumor volume segmentation.

### 2.3. Post-Processing and WM Tract Definition

Motion and eddy current correction on acquired DTI data was performed in eddy [56]. DTI data reconstruction was performed using q-space diffeomorphic reconstruction (QSDR) in DSI Studio (http://dsi-studio.labsolver.org accessed on 18 January 2024) with a sampling length ratio of 1.25 and output resolution of 2 mm. In short, QSDR is a WM-based nonlinear registration approach that reconstructs diffusion information in MNI space [57]. The T2-FLAIR for each patient was included in the reconstruction, thus spatially normalized to MNI space. Diffusion-based parametric images of FA, AD, RD, MD and QA were derived in MNI space.

Five bilateral sets of WM tracts (Frontal Aslant tract, FAT; Arcuate fasciculus, AF; Inferior Fronto-Occipital Fasciculus, IFOF; Cortico-spinal tract, CST; and Cingulum, Ci) were reconstructed based on the HCP-1065 template as previously described [58]. Along-tract analysis was performed for each diffusion-based parameter on all five bilateral WM tracts. Lengthwise mapping was performed using the tract profile function in DSI Studio. In brief, all WM tracts were stretched to correspond straight lines and normalized to a length of 100 segments of arbitrary length. Values derived from the diffusion-based parameter images and the binary segmented tumor image were sampled along the mapped WM tracts and regressed using a kernel density estimator with default regression bandwidth at 1.0.

### 2.4. Neuropsychological Assessment

Patients were assessed by a trained neuropsychologist (1–7 days) prior to surgery as previously published by our group [13]. The neuropsychological assessment included tests of attention and working memory, visual search speed, immediate learning and retrieval (verbal as well as visual), executive functioning, motor speed of the dominant hand and self-reported anxiety and depression. Test scores were adjusted for age and education according to available norms. A domain was considered impaired if the patient displayed neuropsychological symptoms below −1.5 standard deviation (SD).

### 2.5. Analysis and Statistics

Since WM tracts are three-dimensional objects, the maximal tumor presence (MTP) value based on the binary segmented image, will be interpolated and vary between 0 (no tumor) and 1 (only tumor) for a given segment. Thus, the tumor volume (Tv, maximum 100) in arbitrary units is defined as the summation of the tumor presence values for all segments, and tumor extension (Te, maximum 100) is defined as the number of sequential segments with a non-zero tumor presence. A normal dataset was created for each WM tract by calculating average and SD of contralateral nonaffected WM tracts for each diffusion-based parameter, respectively. WM tracts in the affected hemisphere were compared to the normal dataset using a z-test. No mathematical correction was made for multiple comparisons within each WM tract. Instead, ≥10 sequential segments with z-score −1.96 or +1.96, corresponding to a two-sided *p*-value of 0.05, were considered abnormal segments (with or without the presence of tumor).

A graphical presentation was prepared for each WM tract and diffusion-based parameter for all patients, respectively, for visual qualitative assessment. The patient-specific along-tract analysis is plotted with segments (0–100) on the *x*-axis and diffusion-based parameters on the left *y*-axis. Corresponding normal data for the given WM tract are plotted together with SD. Squares at the bottom of the graphs indicate whether significant differences between the patient-specific WM tract and normal data are present for each segment; thus, as described above, ten sequential segments indicate an abnormal segment. Tumor presence is plotted as a light grey shadow using the right *y*-axis.

Based on the graphical presentation, sensitivity and specificity were calculated for each diffusion-based parameter including all WM tracts. Sensitivity was calculated as the number of true positives/(true positives + false negatives) and specificity was calculated as the number of true negatives/(true negatives + false positives). WM tracts with Tv < 1 were considered nonaffected. True positive was defined as abnormal segments, as described above, mostly within the extent of the tumor given on the *x*-axis. True negative was defined as no abnormal segments with no tumor present. False positive was defined as abnormal segments with no tumor present and false negative was defined as no abnormal segments with tumor present. High sensitivity reflects a specific diffusion-based parameters’ ability to correctly identify the presence of a tumor, and high specificity reflects its ability to correctly identify non-tumor presence.

A correlation analysis was performed using Spearman’s rho comparing all diffusion-based parameters and NPS score with Tv, MTP and Te, respectively. Both diffusion-based parameters and NPS score were included in the analysis as binary variables, i.e., abnormal or nonabnormal diffusion-based parameters and impaired or not impaired NPS. Correlation between abnormal diffusion-based parameters and NPS scores was evaluated using Chi-Square test. To understand the role of each variable in respect to neuropsychological impairment, a logistic regression was performed. Univariate analysis included NPS impairment as the dependent variable and diffusion-based parameters, encoded as binary variables (i.e., increased or decreased values compared to normal data), and Tv, MTP and Te as continuous variables. A multivariate logistic regression analysis was performed post hoc only including significant variables in a forward-conditional model to detect independent predictors of neuropsychological impairment.

Derived *p*-values < 0.05 were considered significant. Statistical analysis and graphic design were performed using GraphPad Prism 9 (GraphPad Software, La Jolla, CA, USA) and SPSS 29.0 (SPSS, Inc., Chicago, IL, USA).

## 3. Results

### 3.1. Patient Population

Fourteen patients with IDH-mutated grade 2–3 DGs were included. Eleven patients displayed epilepsy as the onset of symptoms while one patient experienced cognitive impairment, one patient experienced a sensory/haptic symptom as the onset and the last case was an incidental finding. Summary of the population demographics, diagnoses and radiological features is displayed in Table 1. In eleven patients, the tumor was considered eloquent as it involved the minimal common brain as previously described [8,59]. Tumor resection was performed in all patients; resection was performed in the awake condition in six patients and in the asleep condition in eight patients (two patients underwent neurophysiological monitoring).

### 3.2. Sensitivity and Specificity

The highest sensitivities were found in MD (77%) and RD (76%). The sensitivities calculated for FA (70%) and AD (69%) were similar, albeit slightly lower. For QA (30%) a low sensitivity was found. Overall, specificity was found to be equal to or higher than 85% for all diffusion-based parameters. The highest specificities were found in AD (96%) and MD (95%). All values are presented in Table 2.

### 3.3. Tumor Presence

The average tumor volume in arbitrary units per WM tract was 10.7 (SD: 16.4), the average and median maximal tumor presence (MTP) in the affected white matter pathways was 0.28 (median 0.1 ± 0.31) and the average number of tumor segments (Te) along the WM tracts was 31 (SD: 31). A graphical presentation and descriptive values for each WM tract are shown in Figure 1 and summarized in Table 3.

### 3.4. Neuropsychological Results

In twelve patients the neuropsychological assessment was completed before the surgical operation. In two cases the Digit span test was not performed and in one of these cases the RAVLT test results were incomplete. Neuropsychological impairment was detected in six (42%) patients, with two patients presenting impairment in two domains and four patients with only one domain affected. A descriptive summary of the results from the neuropsychological assessment is presented in Table 4.

### 3.5. Correlation Analysis

Moderate significant correlations (rs: 0.70–0.80) were found between the diffusion-based parameters (except QA) Tv, MTP and Te, respectively. Low correlations were found between QA and Tv (*p* = 0.015) and between the NPS scores and all WM-TOP parameters. A Chi-Square test showed that there was no significant association between the AD, MD and RD and NPS scores. The relationship of FA and QA with the NPS scores, respectively, was significant (FA: χ^2^ (1, N = 70) = 4.3, *p* < 0.05, QA: χ^2^ (1, N = 70) = 8.2, *p* < 0.01). The results from the correlation analysis are presented in Table 5.

### 3.6. Regression Analysis

The logistic regression analysis with neuropsychological impairment as the dependent variable showed that abnormal QA was the only independent variable able to predict the functional status of the patients (HR 2.80; *p* < 0.05) (Table 6).

## 4. Discussion

In the current study we used a graphical presentation of segmental along-tract analysis to perform a qualitative analysis of diffusion-based parameters and neuropsychological impairment in patients with DGs.

The visual assessment of WM tracts is made easy with the graphical presentation showing qualitative information and concomitant tumor presence. Similar graphs have also been demonstrated by others [37,55,60] with minor differences. Although the graphical presentation of along-tract analysis may be a potential tool in a clinical workflow, it is highly flawed by the fact that the WM tracts have been reconstructed in MNI space and have been altered length wise, hence they cannot be directly translated to patient-specific cases. Still, one advantage in this case is the ability to perform comparisons with a normal dataset which would otherwise be difficult. Moreover, the direction of anatomical structures may be identified, showing more temporal or frontal abnormalities detected in Figure 1. If implemented, this may be an important piece of information from the neurosurgical perspective (to individualize treatment) or even to plan tailored radiotherapy.

We found that MD has the highest combined sensitivity and specificity to detect DGs in segmental along-tract analysis. Furthermore, all abnormal MD segments showed higher values compared to the normal data, suggesting MD to be a robust metric in line with others [61]. We also demonstrated that QA may be useful in predicting NPS impairment. However, the sensitivity of QA was relatively low, indicating that, in general, it is difficult to detect a tumor using QA as a stand-alone parameter. Celtikci et al. reported that QA may help to differentiate between infiltration and displacement [36]. However, no correlations with single NPS scores were performed and we did not perform an in-depth analysis of their specific correlations with WM tracts because of the small sample size and heterogeneous tumor locations. Furthermore, we are not able to define if the detected abnormalities in our study were related to WM tract being infiltrated, dislocated or a combination thereof, hence our results should be carefully interpreted. We believe that this parameter should be investigated more in future studies on WM–glioma interactions since it seems to be less sensitive to peritumoral edema/damages [27,36].

The displacement, infiltration and disruption of WM tracts secondary to DGs have previously been defined using diffusion-based parameters [62,63]. Of note, the WM tracts used in this study are not tracked individually, but are based on a healthy template within MNI space [58]. Consequently, using these parameters, we would not capture the true trajectory of a displaced WM tract, but a combination of WM tract, displaced tissue and/or tumor depending on the extent of the dislocation. In the case of infiltration, we would measure the WM tract along the abnormal infiltrated segment.

### Limitations and Future Directions

There are several limitations in this study. First, the sample size is relatively small and the tumor locations and volumes are heterogeneous. This may affect the interpretation of the results, especially the correlation analysis and logistic regression analysis which should be carefully interpreted. The sample size is an inherent limitation of patients with DLGGs. In our center, only patients who are eligible for a surgical resection undergo a neuropsychological/cognitive assessment, with only 14 patients that met the inclusion/exclusion criteria stated above. This may represent an additional limitation since this group may not be large enough to validate our results. However, IDH-mutated tumors showed a similar age span, and both in our cases and according to the literature, had a lower tendency to show cognitive impairment at the onset [64]. This may be an important aspect to consider for future studies that will need to confirm these results on a larger and even more heterogeneous population of patients with different types of gliomas.

The number of WM tracts analyzed is also a limitation since the locations are heterogeneous. However, since our aim was an explorative study to investigate diffusion-based parameters in major WM tracts, our results should be considered preliminary even if they confirmed the expected relationship between WM tract location and tumor extension.

Further studies with larger patient cohorts are warranted to confirm our results and to implement the method. One important aspect will be to analyze patients with tumor harboring the same area and compare the results from the same surrounding white matter bundles. Moreover, with a larger sample size one should potentially correlate different regions of specific WM tracts and specific neuropsychological tests, which was not possible in this this study. With respect to diffusion-imaging methodology, it will be important to compare different methods which are now available such as neurite orientation dispersion and density imaging (NODDI) which can address some of the current limitations within DTI [37]. Finally, to confirm the specific and regional infiltration of the WM tracts by tumor cells, an intraoperative validation will be necessary, for instance, to sample different areas of tumor/white matter during operation and analyzing cell density and possible infiltration with separate analyses will give important clinical relevance to this non-invasive method.

Along-tract analysis has the potential to be utilized a tool for the regional non-invasive detection of tumor infiltration to tailor treatment management, such as in precision radiotherapy, or to confirm treatment response in the case of antitumoral drugs, as shown by some recent studies [5].

## 5. Conclusions

The graphical presentation of along-tract analysis presented in this study may be a sensitive and robust method to acquire and display topographical and qualitative information regarding the integrity of WM tracts in close proximity to DGs. Further studies and refinements to the methods presented herein may advance current clinical methods for evaluating displacement and infiltrations and further aid the efforts of pre-planning surgical interventions with the goal of maximizing EOR and individualizing treatment algorithms.

## Figures and Tables

**Figure 1 brainsci-14-00213-f001:**
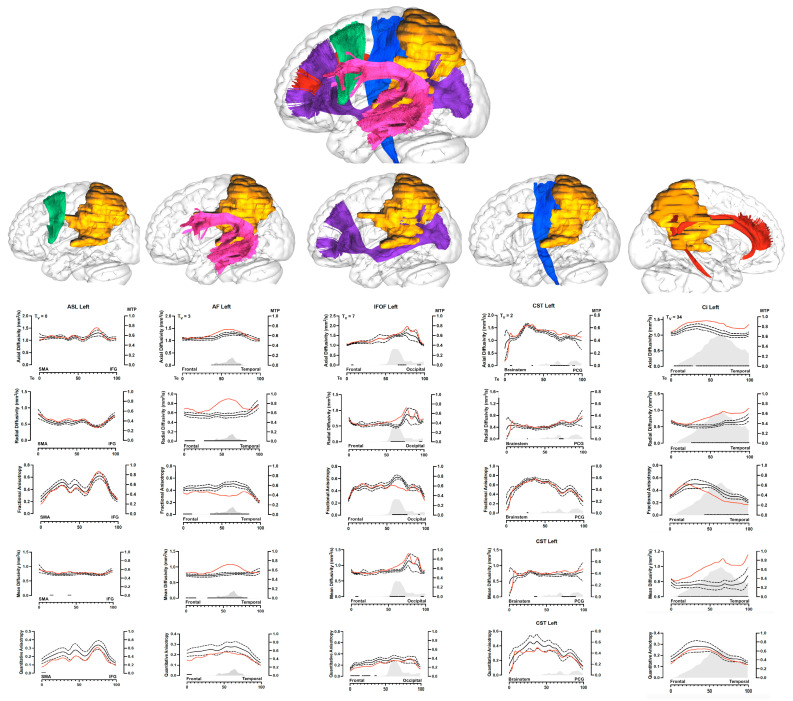
The picture above illustrates the case of patient n°3. The upper part shows a 3D tumor reconstruction (in orange) and its relationship with the five WM pathways; reconstructed with different colors (FAT, green; AF, magenta; IFOF, purple; CST, blue; Ci, dark red) within a 3D “glass-brain” in the left hemisphere. SMA: supplementary motor area; IFG: inferior frontal gyrus; PCG: precentral gyrus. In the lower part each WM pathway is displayed separately in the first row with its relationship with the tumor. The graphs on the second, third and fourth, fifth and sixth rows display the ATA analysis of each WM pathway for all the DTI indices. The red line represents the patient´s white matter index, the continuous black line represents the normal values with standard deviation (black dotted lines). The black dots on the lower part show the areas of significant difference between the patient´s index and the normal values at the z-test. Tumor volume (Tv) is displayed in units on the top left of the first graph. The gray area on the lower part of the graph shows the tumor overlay for each WM pathway. Within each graph the scale on the left shows DTI values, while the scale on the right Y axis shows the maximal tumor presence per unit of white matter (MTP, between 0 and 1). Tumor extension (Te) along the X axis is shown on a scale from 0 to 100. Major lobar or anatomical locations are displayed on the X axis to better locate the DTI index abnormalities and the direction of potential white matter abnormalities.

**Table 1 brainsci-14-00213-t001:** A summary of demographic and radiological features for the study populations. The tumor diagnosis is displayed in accordance with WHO 2021 criteria. The lobar location is indicated according to the predominant infiltration, in case of equal infiltration multiple lobes are indicated. Radiological borders are displayed here in Bulky (B) which indicates sharp borders on FLAIR images, or diffuse (D), irregular or unclear margins.

Pat n°	Age	Gender	Diagnosis	Lobar Location	Side	Tumor Volume (mL)	Radiological Borders
1	26	M	A2	T	L	63.5	B
2	43	M	A2	F	R	42.1	B
3	42	M	A2	P	L	84.2	D
4	44	M	O2	T-I	L	55.1	B
5	23	F	A2	F	R	7.5	B
6	26	M	O2	T	L	31.8	D
7	26	M	A3	F-T-I	R	182.9	D
8	28	M	A2	F-T-I	R	148.5	D
9	39	M	A2	T	L	7.5	B
10	39	F	O2	F	R	15.7	D
11	45	M	A2	F-T-I	L	45.2	B
12	35	M	O2	F-I	R	92.2	D
13	24	M	A3	P	L	56.7	B
14	35	F	A3	F	R	153.3	D

M: Male; F: Female; A: Astrocytoma IDH mutated; O: Oligodendroglioma IDH mutated and 1p19q co-delated; T: temporal lobe; F: frontal lobe; P: parietal lobe; I: Insular.

**Table 2 brainsci-14-00213-t002:** Derived sensitivity and specificity given in percentage for each diffusion-based parameter, respectively.

	AD	FA	MD	QA	RD
Sensitivity [%]	69	70	77	30	76
Specificity [%]	96	85	95	85	92

AD: Axial diffusivity; FA: Fractional anisotropy; MD: mean diffusivity; QA: Quantitative anisotropy; RD: radial diffusivity.

**Table 3 brainsci-14-00213-t003:** The table shows qualitative results from the visual analysis of the graphical presentation. Bold numbers indicate patients with neuropsychological impairment at preoperative assessment. Arrows indicate whether a given diffusion-based parameter is significantly higher (↑) or lower (↓) compared to normal data within tumor presence. Zeroes given in color represent significantly higher (green) or lower (yellow) diffusion-based parameters compared to normal data without or outside of tumor presence. Tumor volume (Tv) is given in arbitrary units and is defined as the area under the tumor shadow in the graphical presentations. Maximal tumor presence (MTP) per unit is encoded between 0 (non-present) and 1 (only tumor). Tumor extension (Te) is defined as the number of sequential segments comprising tumor presence (varies from 0 to 100).

WM Bundles	Pat n°	1	2	3	4	5	6	7	8	9	10	11	12	13	14
**FAT**															
AD		0	↑	0	↓	0	0	↑	↑	0	↑	↓	↑	0	↑
RD		0	↑	0	↑	0	0	↑	↑	0	↑	↑	↑	0	↑
MD		0	↑	0	↑	0	0	↑	↑	0	↑	↑	↑	0	↑
FA		0	0	0	↓	0	0	↓	↓	0	↓	↓	↓	0	↓
QA		0	0	0	↓	0	0	↓	0	0	0	↓	0	0	↑
Tv		0	14	0	0.5	0	0	44	74	0	39	21	15	0	60
MTP		0	0.5	0	0.1	0	0	0.9	1	0	0.9	0.5	0.5	0	0.7
Te		0	45	0	10	0	0	55	100	0	55	55	45	0	100
**IFOF**															
AD		↑	0	↑	0	0	0	↑	↑	0	0	↑	↑	0	↑
RD		↑	0	↑	↑	0	0	↑	↑	↑	0	↑	↑	↑	0
MD		↑	0	↑	↑	0	0	↑	↑	↑	0	↑	↑	↑	↑
FA		↓	0	↓	↓	0	0	↓	↓	↓	0	↓	↓	↓	0
QA		↓	0	0	0	0	↓	↓	0	↑	0	0	↓	0	0
Tv		12	0	3	15	0.5	0.5	57	38	0.5	0	19	30	1	5
MTP		0.5	0	0.4	0.7	0.1	0.1	0.9	0.8	0	0	0.6	0.7	0.1	0.2
Te		45	0	60	50	5	5	100	75	10	0	70	75	15	45
**AF**															
AD		↑	0	↑	0	0	0	↑	0	↑	0	↑	0	↑	↑
RD		↑	0	↑	↑	0	0	↑	↑	0	0	↑	0	↑	↑
MD		↑	0	↑	↑	0	0	↑	↑	0	0	↑	0	↑	↑
FA		0	0	↓	0	0	0	↓	↓	0	0	↓	0	↑	↓
QA		↓	0	0	0	0	0	↓	0	↑	0	0	0	0	↑
Tv		3	0	3	3	0	0	26	15	2	0.5	19	0	15	13
MTP		0.2	0	0.2	0.1	0	0	0.8	0.5	0.1	0.1	0.5	0	0.6	0.6
Te		30	0	35	40	0	0	65	50	30	10	65	0	85	50
**CST**															
AD		0	↑	↑	0	0	0	↑	↓	0	0	0	0	↑	↑
RD		0	↑	0	0	0	↑	↑	0	0	0	0	↑	0	↑
MD		0	↑	↑	0	0	↑	↑	0	0	0	0	0	0	↑
FA		0	↓	0	0	0	↓	↓	0	0	0	↑	↓	0	↓
QA		0	0	0	0	0	0	↓	0	0	0	↑	↓	↑	↑
Tv		0	6	2	0	0	0	29	8	0	0.5	12	0.5	0.5	22
MTP		0	0.4	0.1	0	0	0	0.9	0.5	0	0	0.5	0	0	0.6
Te		0	25	45	0	0	0	60	30	0	10	40	0	0	50
**Ci**															
AD		0	↓	↑	0	0	0	↑	↑	0	0	0	↑	0	↑
RD		0	↑	↑	0	0	0	0	↑	0	0	0	0	0	↑
MD		0	↑	↑	0	0	0	0	↑	0	0	0	0	0	↑
FA		0	↓	↓	0	0	0	↑	↓	0	↓	0	↑	0	↓
QA		↓	0	0	0	0	↑	↓	0	0	0	0	0	0	↑
Tv		0.5	13	34	0	0	3	2	30	0	3	0	0.5	0	38
MTP		0	0.4	0.6	0	0	0.1	0.2	0.7	0	0.1	0	0.1	0	0.6
Te		10	70	85	0	0	40	20	70	0	45	0	10	0	85

AD: Axial diffusivity; FA: Fractional anisotropy; MD: mean diffusivity; QA: Quantitative anisotropy; RD: radial diffusivity.

**Table 4 brainsci-14-00213-t004:** Summary of the results of the neuropsychological assessment for all included patients. Presented values are scaled scores (scaled scores, Ss; or T-scored, Ts) adjusted to available norms; for scaled score, the mean value was 10 ± 3, while for T-scored, the mean value was 50 ± 10.

Pat n°	Lobar Location	Side	NPS Assessment									
			TMT1 (Ss)	TMT2 (Ss)	TMT3 (Ss)	TMT4 (Ss)	TMT5 (Ss)	DIGIT Span (Ss)	RAVLT Total Learning (Ts)	RAVLT Delayed Recall (Ts)	BVMr- Recall (Ts)	Impairment
1	T	L	9	13	13	12	11	13	43	33	34	Y
2	F	R	11	10	13	7	12	10	59	53	58	N
3	P	L	11	9	11	9	11	14	56	64	53	N
4	T-I	L	10	14	15	14	14	17	36	42	64	N
5	F	R	11	10	10	11	12		53	45	53	N
6	T	L	13	13	13	9	12	9	40	41	20	Y
7	F-T-I	R	8	12	10	9	11					N
8	F-T-I	R	9	10	3	4	12	8	47	60	54	Y
9	T	L	9	13	12	10	13	10	40	46	51	N
10	F	R	13	10	13	12	14	11	52	60	57	N
11	F-T-I	L	11	10	13	11	11	4	50	46	55	Y
12	F-I	R	13	13	13	14	13	17	38	34	44	Y
13	P	L	13	9	13	10	13	15	62	57	53	N
14	F	R	4	7	5	8	10	7	63	49	33	Y

**Table 5 brainsci-14-00213-t005:** The table shows the results from the correlation analysis using Spearmen’s rho and Chi-Square test with *p* value significant at <0.05 (*) and confidence interval at 95%.

**Spearman**	** *p* **	**rs**
AD/Tumor volume	* <0.001	0.76
AD/Maximal Tumor Presence	* <0.001	0.76
AD/Tumor extension	* <0.001	0.75
RD/Tumor volume	* <0.001	0.72
RD/Maximal Tumor Presence	* <0.001	0.70
RD/Tumor extension	* <0.001	0.70
MD/Tumor volume	* <0.001	0.78
MD/Maximal Tumor Presence	* <0.001	0.76
MD/Tumor extension	* <0.001	0.80
FA/Tumor volume	* <0.001	0.73
FA/Maximal Tumor Presence	* <0.001	0.72
FA/Tumor extension	* <0.001	0.72
QA/Tumor volume	* 0.015	0.29
QA/Maximal Tumor Presence	0.050	0.26
QA/Tumor extension	0.079	0.21
NPS/Tumor volume	* 0.007	0.33
NPS/Maximal Tumor Presence	* 0.010	0.32
NPS/Tumor extension	* 0.029	0.27
**Chi-Square Test**	** *p* **	**Con. Co**
AD/NPS	0.07	0.22
RD/NPS	0.18	0.16
MD/NPS	0.18	0.16
FA/NPS	* 0.04	0.25
QA/NPS	* 0.01	0.33

Rs: Rho coefficient; Con.Co: contingency coefficient; AD: Axial diffusivity; RD: radial diffusivity; MD: Mean diffusivity; FA: Fractional anisotropy; QA: quantitative anisotropy.

**Table 6 brainsci-14-00213-t006:** The table shows the results from the univariate and multivariate logistic regression performed with neuropsychological (NPS) impairment as the dependent variable and the forward-conditional method. *p* values were significant at <0.05 and Confidence Interval (CI) at 95%.

**Logistic Regression Univariate**	** *p* **	**HR**	**CI (95%)**
NPS/AD	0.356	1.42	0.68–3.00
NPS/RD	0.591	1.21	0.60–2.46
NPS/MD	0.591	1.21	0.60–2.46
NPS/FA	0.277	1.50	0.72–3.11
NPS/QA	* 0.048	2.80	1.01–7.77
NPS/Tumor volume	0.066	1.04	1.00–1.07
NPS/Tumor extension	0.224	1.01	1.00–1.02
**Logistic Regression Multivariate**	** *p* **	**HR**	**CI (95%)**
QA (Abnormal)/NPS	* 0.048	2.80	1.01–7.77

HR: Hazard ratio; AD: Axial diffusivity; RD: radial diffusivity; MD: Mean diffusivity; FA: Fractional anisotropy; QA: quantitative anisotropy. *: significant values with *p* < 0.05.

## Data Availability

Post-processed MRI data from the original study that support the findings are available on request from the corresponding author. The patient-related data are not publicly available due to privacy or ethical restrictions.

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
