# Peer review of "Qualitative and Visual Along-Tract Analysis of Diffusion-Based Parameters in Patients with Diffuse Gliomas"

_brainsci, 2024, doi:10.3390/brainsci14030213_

Round 1
Reviewer 1 Report
Comments and Suggestions for Authors
Abnormalities in DTI parameters were detected in WM pathways with tumor overlay at z-test and merged with an overlay of WM-tumor overlay parameters (WM-TOP). Topographical and quantitative information were 30 displayed within the same graph. AD and MD displayed the highest linear correlation with WM TOP. Abnormal QA displayed a linear correlation with the recorded tumor volume per WM tract. Neuropsychological impairment was correlated with all the WM-TOP and with abnormal FA (p <.05) and abnormal QA (p <.01). At the logistic regression analysis abnormal QA was the only independent variable able to predict the presence of neuropsychological impairment of the patients.
· Only one paragraph in the introduction?
· How do you make sure patients were assessed by a trained neuropsychologist (1-7 days) prior to surgery?
· Give the calculation formula of “sensitivity and specificity were calculated for each diffusion-based parameter including all WM tracts”.
· Can you draw the ROC curve?
· How do you solve the limitation of “sample size is relatively small and tumor locations and volumes are heterogeneous”?
· Section 5, you need to include more future directions.
Author Response
Reviewer 1
Only one paragraph in the introduction?
Answer: We have extended the introduction to better help the reader regarding the patient group, surgical management, DTI description, and neuropsychological-DTI correlation.
How do you make sure patients were assessed by a
trained neuropsychologist (1-7 days) prior to surgery?
Answer: Two trained neuropsychologists have been part of the team since 2016 and they have performed neuropsychological assessment (cognitive and neuropsychological tests) in patients harboring gliomas preoperative and postoperative since 2016 (as demonstrated by other publications of our group). The preoperative assessment with neuropsychologist and speech therapist is an integral part of our clinical routines and has been included as a part of a larger study on intraoperative mapping and higher cognitive functions in patients with gliomas. Moreover, these patients are often scheduled for surgery within 2-6 weeks from the radiological diagnosis. In all the cases we could schedule the preoperative assessment and discuss the results before the operation.
Give the calculation formula of “sensitivity and specificity
were calculated for each diffusion-based parameter
including all WM tracts”
Answer: This has been added in section 2.5 Analysis and statistics
Can you draw the ROC curve?
Answer: This is unfortunately not possible. ROC curves are constructed by varying a threshold for the binary classifier. Since this is a qualitative study where the classification is based on visual assessment this is difficult.
How do you solve the limitation of “sample size is relatively
small and tumor locations and volumes are
heterogeneous”?
Answer: This is a well-directed comment which is however difficult to address. The sample size is indeed small and the sample consists of several heterogeneous factors. However, this is not an issue we can solve. The solution would be to include more patients but there are always difficulties in recruiting patients in single-centre studies where the heterogenous factors typical to the disease are known. On the other hand, we believe that this study may work to add to the proof of concept regarding the ATA analysis. The next step will be to divide our large population by anatomical location and compare patients with similar infiltration patterns. This however will require a longer time to recruit and analyse.
We address this issue by being honest within the manuscript that this is a limitation that may affect the results and conclusions drawn.
Section 5, you need to include more future directions.
Answer: We have now included more aspects into the limitation paragraph with direct links to the future direction and possible application of this method.
Reviewer 2 Report
Comments and Suggestions for Authors
In the manuscript, the authors analyze and compare diffusion tensor tractography (DTI)-based parameters along tracts and variables specific for white matter tumor interaction in 14 patients with diffuse gliomas grade 2-3 and correlate them with preoperative neuropsychological assessment and they find that abnormalities in DTI parameters were detected in white matter (WM) pathways with tumor overlay at z-test and merged with an overlay of WM-tumor overlay parameters (WM-TOP). Axial diffusivity (AD) and mean diffusivity (MD) displayed the highest linear correlation with WM-TOP. Abnormal quantitative anisotropy (QA) displayed a linear correlation with the recorded tumor volume per WM tract. Neuropsychological impairment was correlated with all the WM-TOP and with abnormal fractional anisotropy (FA) and abnormal QA. Abnormal QA was the only independent variable able to predict the presence of neuropsychological impairment of the patients. The topic is interesting, however, some major concerns should be addressed.
1. Why the authors select the patients with IDH-mutated DG grade 2-3? This detection method is not suitable for other type gliomas? What is the inclusion criteria?
2. The sample size is relative small. The statistics results might be not accurate. Please seek advice from a statistician.
3. What does the sensitivity and specificity mean? How to calculate the sensitivity and specificity?
4. The sensitivity of QA is 30%, however, the result demonstrates that this parameter was the only independent variable able to predict the presence of neuropsychological impairment of the patients. What‘s the clinical implication of this parameter?
Author Response
Reviewer 2
Why the authors select the patients with IDH-mutated
DG grade 2-3? This detection method is not suitable
for other type gliomas? What is the inclusion criteria?
Answer: Thanks for this relevant question. We have now specified in the limitation paragraph the reason for the choice. We defined the inclusion criteria in the patient population paragraph. In our center patients with suspected low-grade gliomas suitable for surgical resection (no biopsies cases) are enrolled in a larger study of cognitive functions and surgical aspects. Between 2020 and 2020 only 14 patients fulfilled the criteria of radiological suspected low-grade gliomas, no previous resection for brain tumors, no previous radio-chemotherapy, no background of psychiatric or severe cognitive impairment, and no impossibility to perform neuropsychological assessment due to medical constraints.
We hope that the detection method would be suitable for other types of gliomas but at the moment we do not have enough data to confirm that. Since the group was already heterogeneous and with other limitations (location, volumes, grade 2-3) we decided to analyze a group at least homogeneous on the age and functional level.
The sample size is relative small. The statistics results
might be not accurate. Please seek advice from a
statistician.
Answer: This is a valid comment, and yes, the sample size is small and there are several heterogenous factors that come into play. The statistical analysis was performed on three different levels, on the whole group of patients (14) on each white matter bundle (100 samples) and each DTI parameter for all the white matter bundle (14x5= 70). The major limitation related to the sample size is the correlation with neuropsychology. We have stated this limitation at the end of the discussion and we do not believe that the other level of analysis is affected crucially. This issue with sample size is related to difficulties in recruiting patients with a heterogeneous presenting disease. A larger study cohort based on multicentric recruitment has already begun but the results presented in this study will be used to guide the next level of analysis.
What does the sensitivity and specificity mean? How
to calculate the sensitivity and specificity?
Answer: This has been added in section 2.5 Analysis and Statistics. High sensitivity reflects a specific diffusion-based parameter's ability to correctly identify the presence of a tumor, and high specificity reflects the ability to correctly identify non-tumor presence. This has been added in section 2.5 Analysis and Statistics with how they are calculated.
The sensitivity of QA is 30%, however, the result
demonstrates that this parameter was the only
independent variable able to predict the presence of
neuropsychological impairment of the patients. What‘s
the clinical implication of this parameter?
Answer: Thanks for a relevant comment. There are not so many studies linking QA to neuropsychological/ neurological functions. We do not know. In the discussion, we state that our results from the correlation analysis should be carefully interpreted. Only a study with a larger cohort of patients and more selective analysis on specific white matter bundles and maybe comparing different diffusion methods will clarify this matter. At the moment we do not believe that we have already enough data or strong results to describe a clinical implication of these results. We, on the other hand, believe that this parameter should be investigated more in the future since it seems to be less sensitive to peritumoral edema/damages.
Round 2
Reviewer 2 Report
Comments and Suggestions for Authors
No additional comments.